



# Wave dispersion and dissipation in landfast ice: comparison of observations against models

Joey J. Voermans[1], Qingxiang Liu[2,1], Aleksey Marchenko[3], Jean Rabault[4,5], Kirill Filchuk[6], Ivan Ryzhov[6], Petra Heil[7], Takuji Waseda[8], Takehiko Nose[8], Tsubasa Kodaira[8], Jingkai Li[2], and Alexander V. Babanin[1,9]

[1]Department of Infrastructure Engineering, University of Melbourne, Parkville, Australia
[2]Physical Oceanography Laboratory, Ocean University of China, Qingdao, China
[3]The University Centre in Svalbard, Longyearbyen, Norway
[4]Norwegian Meteorological Institute, Oslo, Norway
[5]Department of Mathematics, University of Oslo, Oslo, Norway
[6]Arctic and Antarctic Research Institute (AARI), St. Petersburg, Russian Federation
[7]Australian Antarctic Division and Australian Antarctic Program Partnership, University of Tasmania, Hobart, Australia
[8]Graduate School of Frontier Sciences, The University of Tokyo, Kashiwa, Chiba, Japan
[9]Laboratory for Regional Oceanography and Numerical Modeling, National Laboratory for Marine Science and Technology, Qingdao, China

**Correspondence:** Joey Voermans (jvoermans@unimelb.edu.au)

**Abstract.** Observations of wave dissipation and dispersion in sea ice are a necessity for the development and validation of wave-ice interaction models. As the composition of the ice layer can be extremely complex, most models treat the ice layer as a continuum with effective, rather than independently measurable, properties. While this provides opportunities to fit the model to observations, it also obscures our understanding of the wave-ice interactive processes, particularly, it hinders our ability to

identify under which environmental conditions these processes are of significance. Here, we aimed to reduce the number of free variables available by studying wave dissipation in landfast ice. That is, in continuous sea ice, such as landfast ice, the effective properties of the continuum ice layer should revert to the material properties of the ice. We present observations of wave dispersion and dissipation from a field experiment on landfast ice in the Arctic and Antarctic. Independent laboratory measurements were performed on sea ice cores from a neighbouring fjord in the Arctic to estimate the ice viscosity. Results

show that the dispersion of waves in landfast ice is well described by theory of a thin elastic plate and such observations could provide an estimate of the elastic modulus of the ice. Observations of wave dissipation in landfast ice are about an order of magnitude larger than in ice floes and broken ice. Comparison of our observations against models suggests that wave dissipation is attributed to the viscous dissipation within the ice layer for short waves only, whereas turbulence generated through the interactions between the ice and waves is the most likely process for the dissipation of wave energy for long periods. The

separation between short and long waves in this context is expected to be determined by the ice thickness through its influence on the lengthening of short waves. Further studies are required to measure turbulence underneath the ice independently of observations of wave attenuation to confirm our interpretation of the results.



## 1 Introduction

When waves propagate from open water into sea ice, their energy decays at a rate as determined by the properties of the sea ice (e.g., Shen, 2019; Squire, 2020). To model the propagation of wave energy into the ice cover of the polar seas, the impact of the ice on the wave energy balance in wave forecasting models is typically formulated in terms of an ice damping source term $S_{ice}$ which, when wave scattering is assumed to be insignificant, is given by (e.g., Shen, 2019):

$$S_{ice} = -\alpha c_g E \tag{1}$$

where $\alpha$ is the apparent spatial attenuation rate, $c_g$ is the group velocity (and can be determined from the wave dispersion relationship) and $E$ is the wave energy density, and both $\alpha$ and $c_g$ are strongly dependent on the local wave and ice properties. Evidently, following Eq. 1, $\alpha$ and $c_g$ are fundamental variables which require parameterisations based, preferably, on the physics that underpin the relevant wave-ice interactive processes.

  Most of the dissipative processes describing the change of wave energy into the ice cover can be organized in two categories:
those that attribute wave energy dissipation to (i) the properties of the ice layer, such as viscous (e.g., Weber, 1987) and visco-elastic theory (e.g., Squire and Allan, 1977; Wang and Shen, 2010), and (ii) viscous or turbulent kinetic energy dissipation in the water surrounding the ice, such as bottom friction (e.g. Kohout et al., 2011; Voermans et al., 2019), overwash (e.g. Toffoli et al., 2015; Nelli et al., 2020) and ice-floe-floe interactions (e.g., Rabault et al., 2019) (we refer to Squire (2020), Shen (2019) and Collins et al. (2017) for a comprehensive overview on wave-ice interaction processes). Each of these processes relies, one
way or another, on the physical and material properties of the ice. Sea ice is, however, a complex medium and can consist of a mixture ice types (e.g., frazil and consolidated ice), different length scales (from pancake ice to a continuous ice sheet), and even within each type the material properties of each element can vary greatly (e.g., first year versus multi-year ice). As each physical and material detail of the ice can have a leading impact on the transformation of the wave field, capturing such variability at global scales for modeling purposes is challenging.

A common approach in tackling this obstacle is by treating the sea ice as a continuum, that is, it is assumed that the ice can be represented by a homogeneous ice layer with 'effective' ice properties rather than measurable material properties. The effective properties are then, ultimately, a function of the ice layer characteristics. Thus, if the effective properties of the ice are known, the development of waves in ice (i.e. $\alpha$ and $c_g$) can be modelled at macroscopic scales. The calibration of these continuum models against in-situ and satellite observations has been the topic of many studies on wave-ice interactions (e.g.,
Ardhuin et al., 2016; Liu et al., 2020). However, a critical problem with this approach is that all models can, to a certain degree, be fitted to the observations even if the physical process upon which the model was founded is of no relevance in the environmental setting. It is perhaps for this reason that there is still a very limited understanding of how much each dissipative process actually contributes to the total dissipation rate under any given ice and wave conditions. Instead, our current practical understanding of the wave attenuation rate tends to be restricted to the power dependency of the wave attenuation rate $\alpha$ with





wave frequency $f^n$, where $n$ tends to vary between 2 and 4 (Meylan et al., 2018) which, in turn, gives us clues which processes could be of importance (Rogers et al., 2021).

Rather than attempting to parameterise the effective properties of the models for different ice conditions, a new perspective on the functioning of models and theories may be provided when near-homogeneous ice conditions are studied (realistically, the ice layer is never be perfectly homogeneous). For example, when a continuous ice sheet is considered, such as landfast ice,

the effective properties of the ice layer as per the continuum model should revert to the material properties of consolidated ice which, theoretically, may be measured independently. This then reduces the number of free variables available to fit models to observations. However, *in situ* observations of wave-ice interactions in landfast ice are rare, perhaps with the notable exceptions of Sutherland et al. (2019) and Kovalev et al. (2020). Therefore, to provide further insights in the wave-ice interactive processes that could play a dominant role in the transformation of waves propagating in sea ice, we performed two field experiments on

landfast ice. Specifically, the ice motion was recorded over the duration of a few weeks on landfast ice in the Arctic and Antarctic. The observations are used to determine the wave attenuation rates and estimate the dispersion relationship in a continuous ice sheet, and the results are compared against available theories and models. To support our comparison against contemporary wave dissipation models, we use estimates of sea ice viscosity obtained from laboratory tests on sea ice cores taken from a neighbouring fjord at the time of our Arctic field experiments.

## 2   Methods

### 2.1   Field Experiments

Two field experiments were performed on landfast ice to measure the wave-induced ice motion, one in the Arctic, and the other in the Antarctic. In both experiments, open-source ice motion loggers were used (Rabault et al., 2020), hereafter referred to as 'ice buoys'. The ice buoys recorded ice motion using a high accuracy inertial motion unit (IMU, VectorNav VN-100) at a

frequency of 10 Hz and transmitted the full wave spectrum, geographical location and battery status, every 2:45 and 4:15 hours for the Arctic and Antarctic experiment, respectively, through Iridium connectivity.

In the Arctic experiment, three ice buoys were deployed along the main axis of Grønfjorden, Svalbard, with the first buoy deployed approximately 500 m from the unbroken ice edge. The other two buoys were deployed 600 and 1400 m from the first buoy. Ice thicknesses of 0.3–0.4 m were measured at the start of the experiment. Instruments were deployed for approximately

two weeks after which they were retrieved. The maximum significant wave height measured (defined as $H_s = 4(\int E(f)df)^{1/2}$) was approximately 10 cm. Timeseries of the significant wave height and peak period are shown for reference in Fig. A1a and is taken from Voermans et al. (2020). The reader is referred to this reference for more details of the Arctic experiment.

In the Antarctic experiment, two ice buoys were deployed on landfast ice north of Casey Station (66.2° S, 110.6° E, see Fig. 1) and positioned about 1.9 km apart. The buoys were deployed in October 2020 and retrieved after 3–4 weeks. During

the experiment, 200–300 km of highly concentrated pack ice and a variable region of about 100 km of loosely packed ice, the Marginal Ice Zone, separated the buoys from the Southern Ocean open water. As a result, only limited wave energy was measured during the experiment with significant wave height up to about $H_s = 3$ cm (Fig. A1b). The ice thickness was 1.1


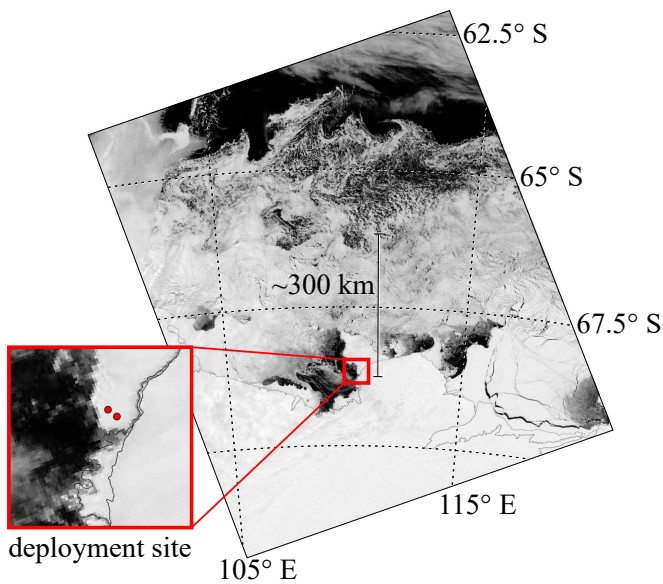

**Figure 1.** MODIS imagery (https://worldview.earthdata.nasa.gov/, last access: 7 May 2021) of the deployment site and sea ice on the 10th of November (end of deployment with clear sky). Instruments were deployed on landfast ice and the site was separated from the Southern Ocean open water by a vast stretch of pack ice throughout the deployment.

m before and 1.3 m after the experiment. The local water depth at the deployment site was unknown but has been estimated, based on the wave dispersion relationship, at about 120 m.

There were no mechanical tests performed on local sea ice in either of the field experiments. However, at the time of the Arctic experiment, mechanical tests were performed through an independent project in a neighbouring fjord to estimate sea ice viscosity. This experiment is summarized in Section 2.3, and, together with observations taken from literature, will serve as a proxy of the ice viscosity used in the comparison against various wave attenuation models. The methods and results of the mechanical tests and estimation of the ice viscosity are provided in Appendix B.

**2.2    Data Processing**

Heave spectra were derived from the vertical acceleration spectra $E_A(f)$ as per $E(f) = 1/\omega^4 E_A(f)$. Examples of spectra are shown in Fig. A2. To avoid spurious results by instrument noise we use here a signal-to-noise-ratio SNR $\geq 2$ as threshold of acceptable data (e.g., Thomson et al., 2021). The noise level of the IMU was determined by fitting a power relationship through the high frequency range of the spectrum $E(f)$ where no wave energy is expected nor observed. The noise level is
then removed from the spectra.





To determine the wave dispersion relation in ice, the wave number is estimated from the measurements of heave, pitch and roll:

$$k(f) \approx \sqrt{\frac{E_\alpha(f) + E_\beta(f)}{E(f)}} \tag{2}$$

where $E_\alpha$ and $E_\beta$ are the spectra of the roll and pitch motion, respectively (Kuik et al., 1988; Collins et al., 2018). The Arctic
experiment provided 110 estimates of $k(f)$. Similar to Collins et al. (2018), we notice an average bias of approximately 3% for $k/k_{ow}$ for the lower wave frequencies, e.g., for wave periods between 8 and 12 s, where $k_{ow}$ is the wave number in open water. As the observed wave energy in the Antarctic experiment is significantly smaller than in the Arctic experiment, fewer data passed the SNR threshold criterion. A total of 38 spectra passed quality control, however, the number of valid observations within individual frequency bands varied between 3 and 19.

To estimate the dissipation of wave energy by sea ice, we assume that the spatial dissipation rate is well described by an exponential function (e.g., Wadhams et al., 1988):

$$E(f,x) = E(f,0)e^{-\alpha x} \tag{3}$$

where $x$ is the distance of wave propagation into the ice pack. The buoys did not measure always at the same time due to variable quality of satellite connectivity causing a drift in the starting time of each record. For this analysis we therefore only
consider data pairs obtained within $\Delta t = 30$ minutes of each other. In Eq. 3 it is assumed that the direction of wave propagation is aligned with the axis of the buoy pair. In case of the Arctic experiment, this seems a reasonable assumption as the buoys are aligned with the main axis of the fjord. For the Antarctic experiment, this assumption was tested using ERA5 re-analysis data in the open water just north of the marginal ice zone indicating a relative bearing of approximately 15°. We further remove records where more than 25% of the frequency bands have a negative attenuation rate. Implementing these two additional
criteria leaves 9 profiles of the wave attenuation rate for the Arctic experiment, and just 2 for the Antarctic experiment. We note that for the Arctic experiment only those observations are used that were obtained from the buoy pair furthest apart as they were deemed most accurate.

### 2.3 Ice Viscosity

To compare our observations of wave attenuation against visco-elastic models, viscosity input is required. As no straightforward
method is available to measure the viscosity of solid ice through field or laboratory experiments, the material is often simplified as a spring-dashpot model through which the viscosity can be estimated by stress-strain tests on a material sample. Estimates of the ice viscosity are, nevertheless, extremely rare. Tabata (1958) and Lindgren (1986) estimated an ice viscosity of $10^{13}$ Pa·s (for sea ice at -10°C) and $6 \times 10^{10}$ Pa·s (freshwater ice at about -5°C), respectively, whereas more recently the viscosity of laboratory grown solid (salt water) ice was estimated to vary between $10^8$ and $10^9$ Pa·s (Marchenko et al., 2020, 2021).
While it is outside the scope of this study to provide a review on this topic, it is important to stress that different spring-dashpot models may produce different estimates of the ice viscosity.

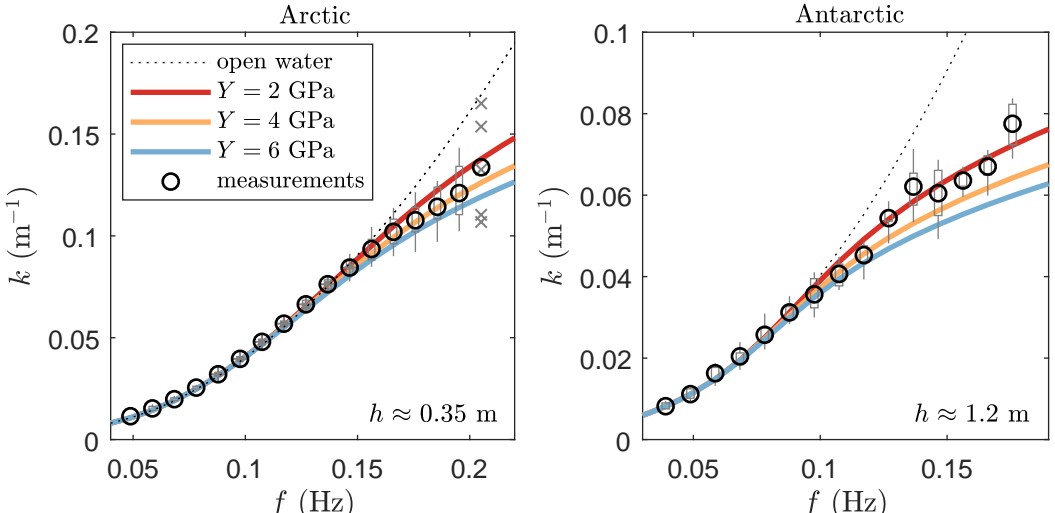

**Figure 2.** Mean values of the estimated wavenumber $k$ in landfast ice based on experimental observations (circle) in the (a) Arctic and (b) Antarctic. Solutions of the dispersion relation following an elastic thin plate are provided in colour. Note that for the Arctic experiment there is only a limited number of observations available at $f = 0.2$ Hz (cross).

In March 2020, a separate field measurement campaign was performed in the Van Mijen Fjord, Svalbard, to measure and estimate the material properties of naturally grown sea ice. Importantly, these measurements were done during the same period as the Arctic field experiment performed as described in Section 2.1, albeit, in a different fjord in Svalbard. In addition to *in situ* cantilever experiments, vertical and horizontal ice cores were taken and tested in the laboratory. Distinction was made between columnar and sea spray ice. By describing the ice as a Burgers material (Maxwell-Voigt spring-dashpot model), the viscosity coefficients were obtained through the estimation of the stress relaxation and elastic lag time scales (Marchenko et al., 2020, 2021). The estimate of the solid ice viscosity was $\mu_i = 3.2 \times 10^{10}$ Pa·s and $3.9 \times 10^{10}$ Pa·s for the columnar sea ice cores and $3.0 \times 10^{10}$ Pa·s for the sea spray ice cores (note that the kinematic viscosity $\upsilon$ is related to the dynamic viscosity as $\upsilon = \mu/\rho$). For clarity, we use subscripts $i$ and $w$ to denote ice and water variables, respectively. Details on this approach and the coefficients obtained from the tests are provided in Appendix B.

Even though the (few available) estimates of consolidated ice viscosity vary by 5 orders of magnitude, $10^8 - 10^{13}$ Pa·s, it does provide critical insight in the approximate bounds for the viscosity of consolidated ice. In this study, we will use this range of $\mu_i$ to study the performance of visco-elastic models against our observations.

## 3 Results

The estimated dispersion relation for each experiment is shown in Figure 2. By averaging in time and across instruments we inherently assume that the ice conditions were constant across the deployment sites and remained unchanged over the duration





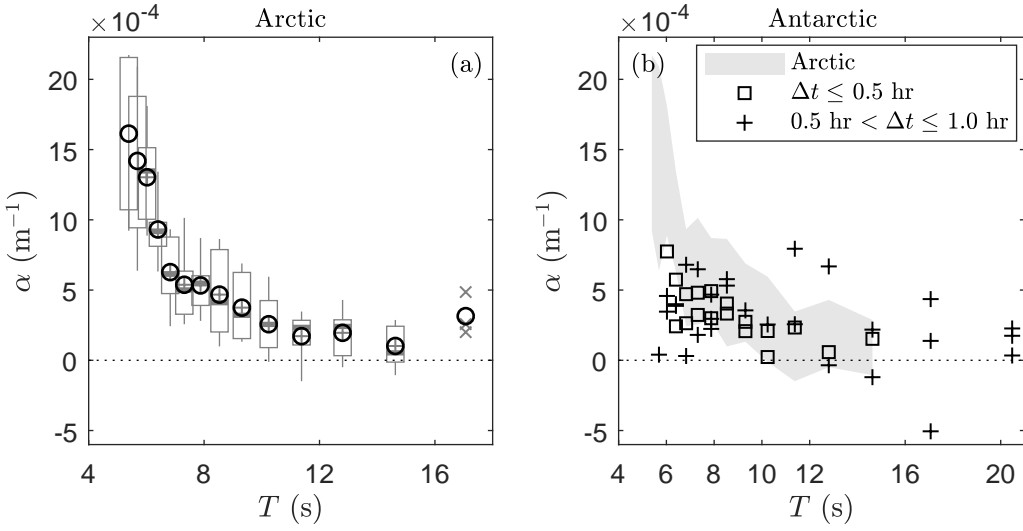

**Figure 3.** (a) Mean wave attenuation rate $\alpha$ as observed in the Arctic (circle). Note that the number of observations for $T = 17$ s in the Arctic experiment (cross) is limited. (b) Observations of the wave attenuation rate in the Antarctic with time difference between co-located buoy measurements below 0.5 hr (square) and between 0.5 and 1.0 hour (plus). Given the limited number of observations, no mean values are provided for the Antarctic experiment, however, the magnitude of the observations corresponds well with those observed in the Arctic.

of the experiments. A lengthening of the short waves is observed in both experiments. The consistent deviation from the open water dispersion relationship for the short waves suggests that the ice was indeed continuous and not broken (Sutherland and

Rabault, 2016).

Observations are compared against the modeled dispersion relationship of a thin elastic plate for different values of the elastic modulus $Y$ (Eq. C7, Liu and Mollo-Christensen, 1988). For the Arctic experiment, the average of $k$ estimates correspond well to the modeled dispersion relation with $Y = 4 \times 10^9$ Pa. For $f \approx 0.2$ Hz, the average value of $k$ deviates from this line yet is well within the range of uncertainty given the limited number of observations at this frequency. Although this value $Y$ is about twice

as large as that estimated based on the measured ice salinity, water and air temperature (Voermans et al., 2020), it is well within the general range of uncertainty of observations of $Y$ (Timco and Weeks, 2010). For the Antarctic experiment, considerably less observations are available resulting in larger scatter of the mean estimate of $k$. Nevertheless, observations correspond reasonably well to the modeled relationship using $Y \approx 2.5 \times 10^9$ Pa. As the ice thickness in the Antarctic experiment is 3–4 times larger than in the Arctic experiment, the lengthening of waves in the Antarctic experiment seems to become significant

at about $f \gtrsim 0.1$ Hz, while at $f \gtrsim 0.16$ Hz in the Arctic experiment.

The mean wave attenuation rate observed in the Arctic experiment decays with an increase in wave period (Fig. 3a) (except for $T = 17$ s, which was discarded from further analysis due to limited data points). As only two wave attenuation profiles of our Antarctic experiment passed the quality control criteria, as outlined in Section 2.2, there are insufficient data available to reliably determine a mean wave attenuation rate for this experiment. Nevertheless, these observations are presented here as the



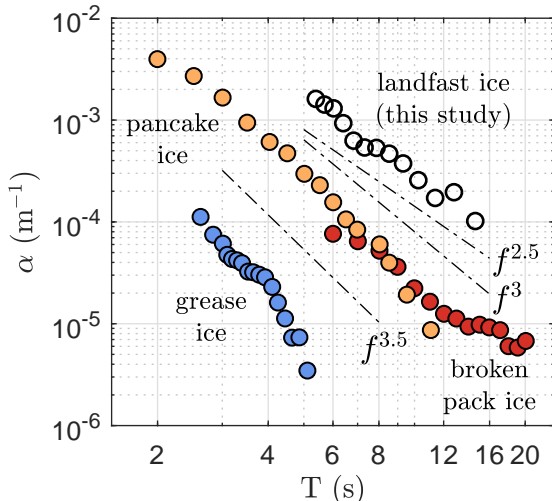

**Figure 4.** Comparison of experiment-averaged wave attenuation rates $\alpha$ for different types of sea ice (all field observations): grease ice (Kodaira et al., 2020), broken pack ice (Kohout and Williams, 2013), pancake ice (Thomson et al., 2018) and landfast ice (this study). Mean attenuation values of the experiments of Kohout and Williams (2013) and Thomson et al. (2018) are taken from Liu et al. (2020). Observations of $\alpha$ in landfast ice fall between $f^{2.5}$ and $f^3$.

little data available do seem to suggest that the wave attenuation rate is of the same order of magnitude as that observed in our Arctic experiment (Fig. 3b). Clearly, loosening the quality control criterion of $\Delta t \leq 0.5$ hr to 1 hr greatly increases the scatter and thus the uncertainty of the wave attenuation observations.

As suggested by Rogers et al. (2016) and Shen (2019), the ice type (whether at micro or macroscopic level) is perhaps the main determinant of the wave energy dissipation rate by the action of sea ice. In Fig. 4 our observations of wave dissipation
in landfast ice (Arctic experiment only) are compared against those observed in other ice types including grease ice (Kodaira et al., 2020), pancake ice (Thomson et al., 2018) and pack ice (Kohout and Williams, 2013). We note that all observations in Fig. 4 are averages per experiment. While within each experiment the magnitude can vary, the average values observed in pack ice are an order of magnitude smaller than what we observe in landfast ice for $5 \leq T \leq 16$. Such difference was also observed by Collins III et al. (2015) and Ardhuin et al. (2020) who suggest that the wave attenuation is dominated by whether the ice is
broken or unbroken. For our observations of $\alpha$ in landfast ice we observe a power regime between $f^{2.5}$ and $f^3$.

In Figure 5 our Arctic wave attenuation observations are compared against different wave dissipation models (an overview of the models is provided in Appendix C). Two processes are considered here: attenuation attributed to the ice layer (cool color tones in Fig. 5), and to the water body (warm color tones). To compare our observations against visco-elastic models, both the elastic modulus and viscosity of the ice need to be known. While we have an estimate of the elastic modulus of ice
during the experiment (i.e., Fig. 2a), the viscosity is unknown. We compare our observations against two different visco-elastic models, the first is that of Squire and Allan (1977), simplified to a Voigt model (Li et al., 2015; Sree et al., 2018). The lower





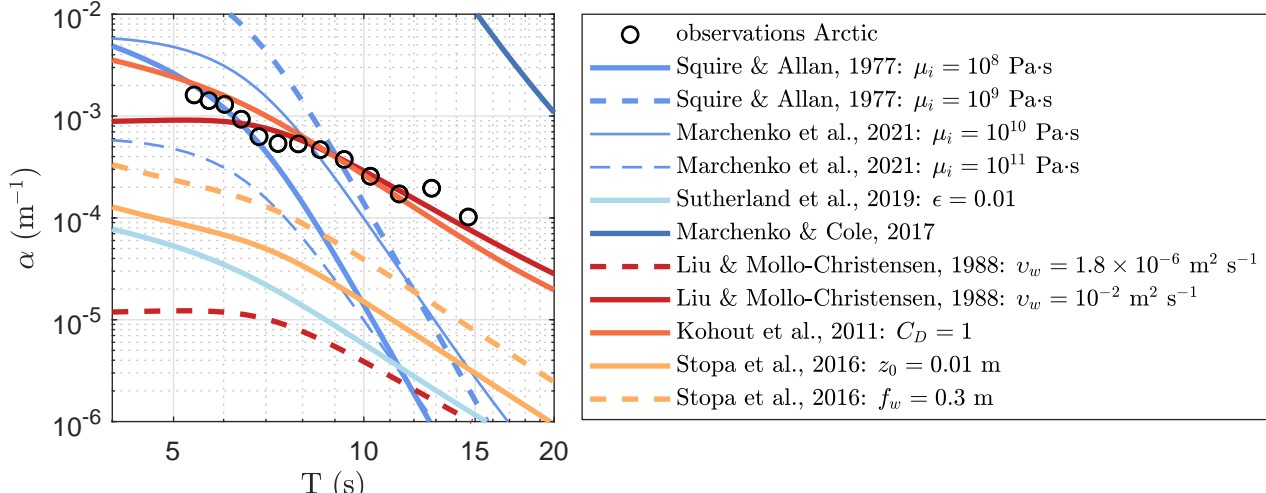

**Figure 5.** Comparison of wave attenuation rate $\alpha$ observed in the Arctic experiment against various wave dissipation models. These include models where dissipation is attributed to the ice layer (cool colour tones, Squire and Allan, 1977; Sutherland et al., 2019; Marchenko and Cole, 2017), and friction (warm colour tones, Liu and Mollo-Christensen, 1988; Kohout et al., 2011; Stopa et al., 2016).

bound value of the ice viscosity is used here, $\mu_i \in [10^8, 10^9]$, to match the observations. We note that this model provides near identical results for the ice parameters used in this study to that of Wang and Shen (2010). For $T < 7$ s, the visco-elastic model provides attenuation estimates of the same order of magnitude as our observations, however, for longer wave periods, there is

a significant discrepancy in both slope and magnitude. The second visco-elastic model used is that of Marchenko et al. (2020, 2021) which is based on a linear Maxwell-Voigt model. The attenuation rates behave similarly as the model of Squire and Allan (1977), but with significantly different order of magnitude. Unlike the visco-elastic model of Squire and Allan (1977), a viscosity value of $\mu_i \in [10^{10}, 10^{11}]$ is required to match the short wave period observations of wave attenuation. We note, that this viscosity range corresponds very well to the ice viscosity estimated through independent ice tests in a neighboring

fjord (see Section 2.3). Fundamentally different from the model of Squire and Allan (1977), is that the model of Marchenko et al. (2020, 2021) is inversely proportional to the ice viscosity, that is, the wave attenuation decreases with an increase in ice viscosity. For the two-layer model of Sutherland et al. (2019) we have assumed a no-slip condition $\Delta_0 = 1$ at the water-ice interface. While there is no physical guidance as to what $\epsilon$ (the relative thickness of the wave permitting ice layer) should be for a solid ice cover, we expect this to be small and thus adopted a value of $\epsilon = 0.01$. This leads to a strongly underestimated

attenuation rate compared to the observed attenuation. The third model related to the ice layer properties is that of Marchenko and Cole (2017) which considers that wave energy is dissipated by brine migration induced by the flexural motion of the ice layer. As supported by the measurements of Golden et al. (2007), sea ice can be considered permeable when ice temperature is greater than about -5°C and thus, the pumping of brine due to flexural deformations of ice is possible only near the bottom





of the ice where the ice temperature is relatively high. This model, however, leads to an overestimation of the attenuation rate

and a much stronger dependency of $\alpha$ on the wave frequency $f$.

The weakest form of wave attenuation by under-ice friction is through the development of a laminar boundary layer under the ice. Comparison against the model of Liu and Mollo-Christensen (1988), with $\upsilon_w = 1.8 \times 10^{-6}$, shows an underestimation of the dissipation rate by two orders of magnitude. To match the observations, the viscosity of the water needs to be increased to an effective value of $\upsilon_w = 10^{-2}$ m$^2$ s$^{-1}$. While the slope reasonably matches the observations for $8 < T < 15$, the attenuation rate

is underestimated for shorter wave periods. Perhaps a more sophisticated way to include the effects of turbulence dissipation is through the model of Stopa et al. (2016), which is based on the analogy with the wave bottom boundary layer. However, with an arbitrary chosen roughness height of $z_0 = 0.01$, the modeled attenuation rate underestimates the observed dissipation rate considerably. Even increasing the friction factor to the limit value of $f_w = 0.3$ still shows an underestimation by an order of magnitude. The last model evaluated is that of Kohout et al. (2011) where the friction at the wave-ice interface is defined by a

drag coefficient. To match the observations, we use a $C_D = 1$. While the fit is reasonable, the model of Kohout et al. (2011) has a slightly larger slope than that observed in the Arctic experiment but matches the observations well across the whole range of observed wave periods.

## 4  Discussion

Our observations of wave attenuation in landfast ice were compared against a variety of models. We find that visco-elastic

theory cannot explain the attenuation rates observed in our Arctic experiment completely. Specifically, the power dependence of $\alpha$ to wave frequency is greatly overestimated for long wave periods (e.g., Meylan et al., 2018; Liu et al., 2020). However, for the shortest waves, the trend and magnitude tend to align well with the observations.

The model that performs comparatively well is the laminar boundary layer model of Liu and Mollo-Christensen (1988) using an effective viscosity of $10^{-2}$ m$^2$s$^{-1}$. If correct, this would imply that the boundary layer under the ice is fully turbulent rather

than laminar. Unlike our observations, the model of Liu and Mollo-Christensen (1988) shows a flattening of the attenuation rate for high frequencies, typically referred to as the rollover effect and attributed to local wind-input and/or non-linear energy transfer (e.g., Li et al., 2017; Thomson et al., 2021). However, given that the ice cover is continuous here and the distance between the ice buoys is relatively small compared to the typical wave length, we do not expect any roll-over effect in our experiments. While the increase of $\upsilon_w$ seems extraordinary, McPhee and Martinson (1994) and Marchenko et al. (2017) did

observe, based on measurements under drifting ice, a similar eddy viscosity of $\upsilon_t = O(10^{-2})$. Yet, in those experiments, it is more likely that the observed turbulence was generated through the relative velocity between the drifting ice and the water rather than the wave orbital motions. Specifically, if one would estimate the magnitude of $\upsilon_w$ during our Arctic experiment as being the product of the orbital wave motion, $O(0.01$ m s$^{-1})$, and the wave boundary layer thickness, $O(0.01$ m) m (that is, the product of the velocity scale and length scale of the largest turbulent eddies generated in the wave-boundary layer) we

would expect a maximum eddy viscosity of $\upsilon_t = O(10^{-4}$ m$^2$s$^{-1})$ instead. Nevertheless, the presence of more complex under-ice roughness, such as ice ridges and platelet ice, could significantly increase the eddy viscosity through increased under-ice



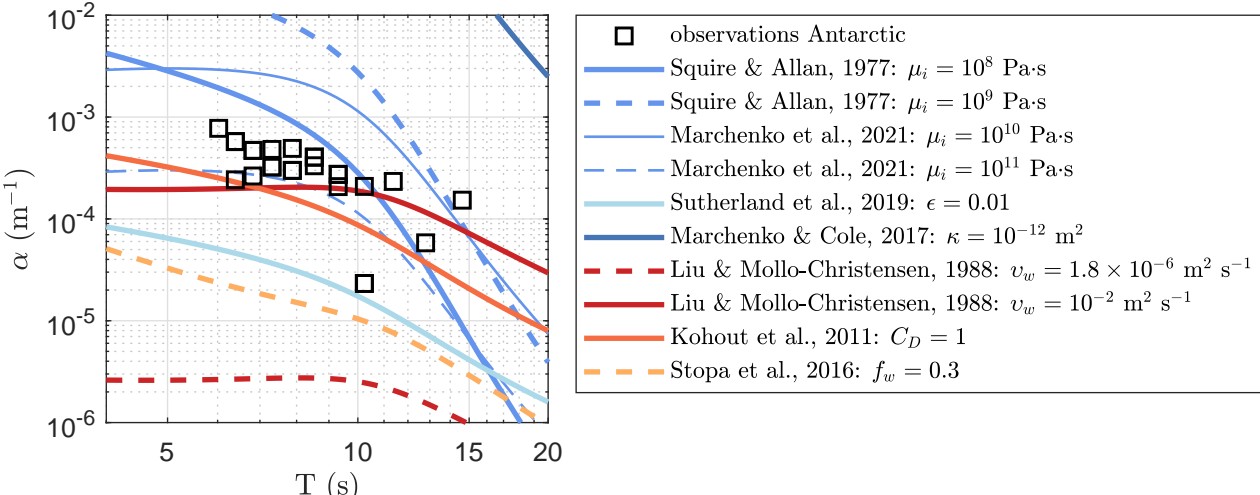

**Figure 6.** Same as Fig. 5, except for the Antarctic experiment. Models have not been fitted to the data. Note, limited observations are available for the Antarctic experiment and results should be interpreted by their order of magnitude rather than fine scale details.

surface roughness. However, without in situ observations of such features, the 'true' eddy viscosity remains uncertain during our experiment.

The relatively simple model of Kohout et al. (2011) works reasonably well across the whole range of observed frequencies

if a drag coefficient of $C_D = 1$ is used. Observations of the under-ice drag coefficient are rare and those reported are often related to currents rather than waves. For currents under the ice, the largest values of the drag coefficient reported are $C_D = O(0.01-0.1)$ (Lu et al., 2011). However, when considering waves rather than currents, $C_D = 4$ has been considered by Herman et al. (2019), whereas Voermans et al. (2019) suggests that the drag coefficient increases exponentially with ice concentration. Extrapolation of their observations to a continuous ice cover gives $C_D = O(1-10)$ which, though highly speculative, makes

a $C_D = 1$ here not implausible.

A critical difference between the dissipative processes of the viscous ice layer and friction at the water-ice interface is that, to the first order, the former depends on ice thickness and not on the local wave height, whereas the latter scales with the wave height while only weakly dependent on the ice thickness. Ice thickness is a second order effect for wave attenuation by boundary layer turbulence as transfer of momentum to the ice is fundamentally determined by the surface roughness properties

of the ice and the hydrodynamics below the ice. Only when second order effects are considered, such as the impact of ice on the dispersion relation and a possible correlation between under ice roughness and ice thickness, the ice thickness can impact the wave attenuation rate by friction. Considering the first order effects to be dominant, the estimated wave attenuation rates from the Antarctic experiment can provide some insight in the relative importance of the two dissipative processes as the ice thickness was 3–4 larger whereas the maximum wave height was 2–3 times smaller than in the Arctic experiment. In Fig. 6

the Antarctic observations are compared against the same models, except here $Y = 2.5 \times 10^9$ Pa, $h = 1.2$ m and $H_s \approx 0.03$ m,





while assuming that the viscosity of the ice and under-ice drag coefficient remain the same. It is important to stress that the comparison of the Antarctic observations of $\alpha$ against the models is speculative, as we have insufficient observations available to determine the mean wave attenuation rate for this experiment. If one would assume the limited Antarctic observations to be representative for the experiment, the visco-elastic model of Squire and Allan (1977) now overestimates the observations by a factor of 3, whereas the boundary layer model of Kohout et al. (2011) underestimates the observations by a factor of 2. The visco-elastic model of Marchenko et al. (2020, 2021) captures the Antarctic field observations well, even if the same ice viscosity values are used as those derived in the Arctic. Unlike the model of Squire and Allan (1977), the attenuation rates of the model of Marchenko et al. (2020, 2021) become constant for $T < 7$ s under the ice conditions in the Antarctic experiment. Thus, taking into consideration the uncertainty in parameterisations and the observations, both under-ice friction and visco-elastic theory could explain the wave dissipation here.

Even though the friction models alone can reasonably replicate the observations of attenuation in both experiments, it is very plausible that both processes are of importance, albeit, at different frequency ranges. The point at which the effective viscosity model of Liu and Mollo-Christensen (1988) and Kohout et al. (2011) starts to flatten, around $6 \leq T \leq 7$, tends to corresponds to that where the visco-elastic model of Squire and Allan (1977) starts to become of comparable magnitude and slope (Fig. 5). A similar observation could be made based on Figure 6 although such a transition would be at a larger wave period of $T \approx 10$ s. This would support the observations of Meylan et al. (2018), who argued that there are two dissipative processes of importance, with one dominating for short waves, the other for long waves. Here, these processes are dissipation due to the ice layer and dissipation beneath the sea ice, respectively. The position of transition observed here seems to be determined by the ice thickness through the wave dispersion relationship by the lengthening of the short waves. This point is likely defined by the frequency at which the elastic effects of the ice dominate the modification of the wave speed in the ice (Fox and Haskell, 2001; Collins et al., 2017). A correlation between wave attenuation rates and ice thickness was also observed by Doble et al. (2015) and Rogers et al. (2021), for pancake and broken pack ice, and considered by Yu et al. (2019) more generally for both wave dispersion and dissipation.

Though the model of Sutherland et al. (2019) significantly underestimates the observations in both experiments, by increasing wave permitting layer to $\epsilon \approx 0.5$ for the Arctic experiment, and $\epsilon \approx 0.1$ for the Antarctic experiment the model results fits well to our observations. This raises questions about the physical interpretation of $\epsilon$ which, similar to the viscous ice models, uses an effective viscosity parameter to capture ice-induced wave dissipation. For a solid and continuous ice layer, the interpretation becomes difficult. Sutherland et al. (2019) reasons that it is possible that sea ice permeability allows wave-induced pressure gradients within a layer $\epsilon$ of the sea ice. Indeed, boundary permeability allows flow penetration across the water-ice interface much like that of coherent turbulent structures at the permeable sediment-water interface (e.g., Voermans et al., 2018) and for large permeability and/or flow induced shear, boundary permeability can lead to significantly enhanced momentum exchange across the interface and thus an enhanced the drag coefficient (or equivalent, the effective viscosity). However, even for a relatively large sea ice permeability of $K = O(10^{-10}\text{m}^2)$ (e.g., Golden et al., 2007), no significantly enhanced momentum exchange is expected if one would assume the analogy with the sediment-water interface to be valid (Voermans et al., 2018). That is, the pressure gradients induced by the waves propagating under the ice is too weak to drive flow within the porous ice





as the resistance of the ice is simply too large. A dissipation process related to ice permeability that might be more plausible is that of brine migration driven by pressure gradients within the ice layer induced by the flexural response of the ice to the waves (Marchenko and Cole, 2017). However, the complexity of this process makes it difficult to identify whether wave energy dissipation by brine migration is an important process in our experiments.

While our observations tend to indicate that boundary layer turbulence and visco-elastic dissipation are dominant dissipative processes in landfast ice in different frequency ranges, one can only be certain of their importance when the mechanical and physical properties of the ice (including the under-ice topography) and details of the turbulent boundary layer under the ice are known. Measuring turbulence under the ice *in situ* is a complex task, not only because of the challenging environment, but also as the thickness of the wave boundary layer is expected to be small, that is, in landfast ice the boundary layer is expected

to be just a few centimeters thick if no extreme roughness formations (such as platelet ice and ice ridges) are present. This complicates the use of acoustic measurement techniques which pose limitations near boundaries due to reflection and low flow velocities, and perhaps optical methods using ROV's could provide a solution to this experimental problem (Løken et al., 2021). As opposed to the need to measure the elastic modulus of the ice through mechanical tests, a potential method to identify the elastic modulus in sea ice, at least for a continuous ice sheet, is by estimating wave number through measurements of heave,

pitch and roll (e.g., see Fig. 2). However, independent mechanical tests still need to verify the accuracy of such a method.

## 5  Conclusions

Observations of waves in landfast ice were used to gain insight into the importance and relevance of various wave-ice interactive processes in the attenuation of wave energy in sea ice. Our estimates of the wave number suggests that the dispersion relation in landfast ice is well described by the thin elastic plate assumption. The ability to estimate the wave dispersion relation using

a single ice buoy implies that the elastic modulus of landfast ice may be estimated using wave observations. We observe wave attenuation rates in landfast ice to be an order of magnitude larger than in broken ice. This is consistent with current understanding that the wave attenuation rate in sea ice is dominated by whether the ice is broken or unbroken, and is strongly determined by the type of sea ice. Results suggest that visco-elastic theory can only explain the observed attenuation rates in landfast ice for short waves whereas the long wave attenuation rates (and in part the short waves as well) are well described by

turbulence and friction based dissipation models. The wave period describing this transition between short and long waves is expected to be dominated by the thickness of the ice. A note of caution is that to match the observations to the turbulence and friction models, large momentum transfer variables are required which, under the experimental conditions, remain physically unexplained. More comprehensive studies are required to substantiate our conclusions by measuring the local physical and material properties of the ice and flow properties underneath the ice independently, particularly the properties of turbulence.

*Data availability.*  Data will be made available in a public repository.





# Appendix A: Wave measurements

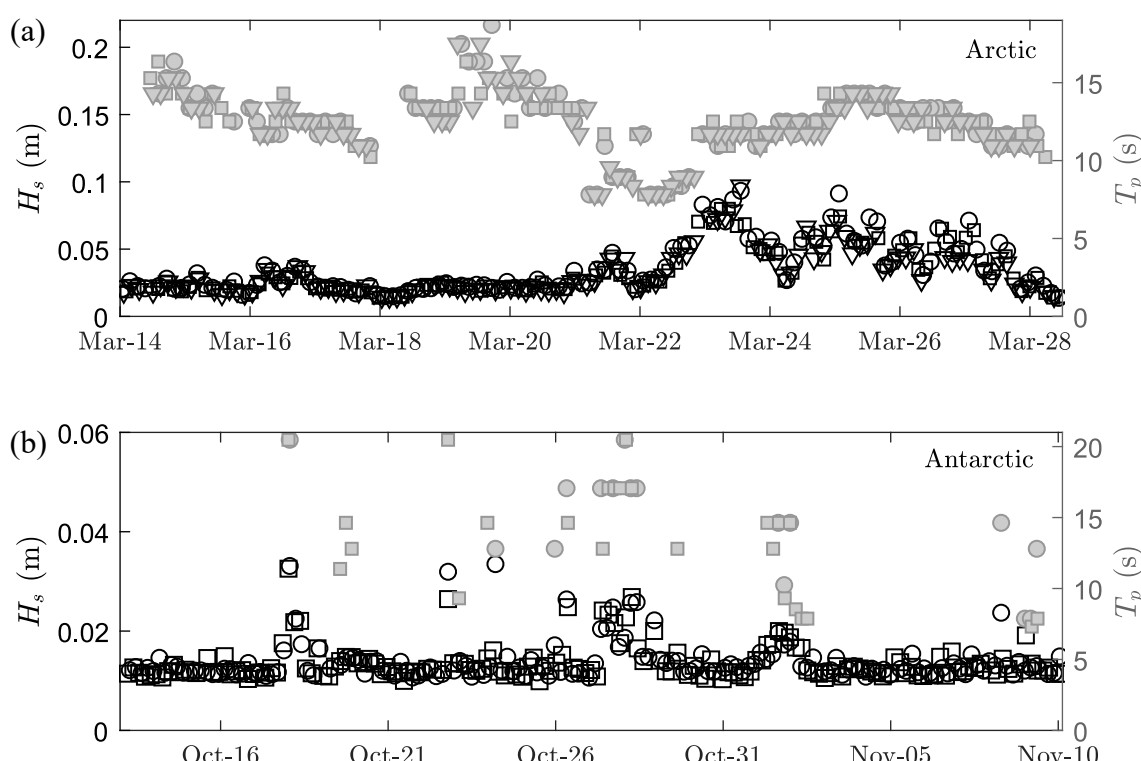

**Figure A1.** Measured wave height (black) and peak period (gray) during (a) the Arctic experiment (taken from Voermans et al., 2020), and (b) the Antarctic experiment. Note that noise threshold of $H_s$ is approximately 1.5 cm. Squares, circles and inverted triangles relate to different instruments within each experiment.





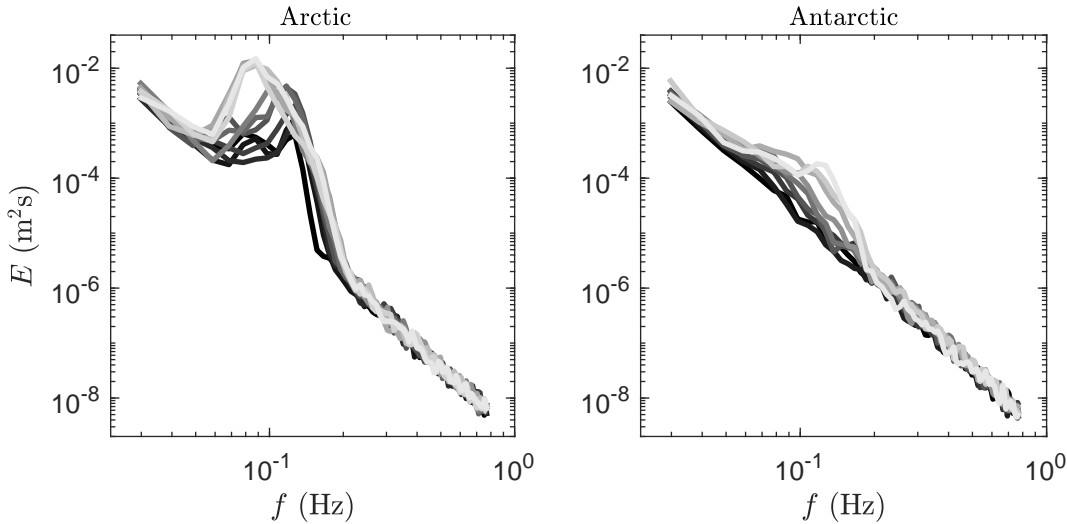

**Figure A2.** Example spectra derived from the vertical acceleration measured by the ice buoys. For the Arctic, the spectra are from 22 March 2020 (black) – 23 March 2020 (light gray). For the Antarctic, the spectra are from 31 October 2020 (black) – 2 November 2020 (light gray). For the Antarctic, only a few spectra pass the SNR criterion.

## Appendix B: Ice Viscosity

### B1 Laboratory Experiments

Spray ice is formed in the coastal zone near Longyearbyen due to regular floods of water onshore due to semidiurnal tide and

waves. The tide height changes between 1 m and 2 m depending on the moon phase. Accordingly, the water line is moving over the coastal slope and freezes. In addition, sea spray freezes along the coastal slope where it accumulates. As a result, a layer of spray ice with thickness 1.5 m was formed along the shoreline to the end of February 2020. The salinity of spray ice was measured from 3.5 to 5.6 ppt. The photographs of vertical and horizontal sections of the spray ice made in polarized light are shown in Fig. B1. The length scale at the left side of the photographs shows length in cm. The ice has very fine granular

structures with maximal grain diameter of about 1 mm.

Winter 2020 was very cold in Spitsbergen, and the thickness of sea ice reached 1 m in the Van Mijen Fjord near Kapp Amsterdam in March 2020. This ice has columnar structure $S_3$ with alignment of c-axes in onshore direction and elongation of the columnar grains in alongshore direction. The photographs of vertical and horizontal sections of the sea ice made in polarized light are shown in Fig. B2. Yellow strip at the left side of the photographs has length 5 cm. The size of columnar

grains in onshore direction is about 2 cm, and the size of columnar grains in alongshore direction is of about 5 cm. The sea ice salinity was measured in the range of 4–6 ppt.

Ice cores taken from spray sea ice and columnar sea ice were used in the laboratory tests on uniaxial compression to calculate elastic and viscous properties of sea ice Fig. B3a. The length and the diameter of ice cores were respectively 175 mm and 72 mm. In each tests ice cores were subjected to constant compressive load over some time and then unloaded (LU test). The



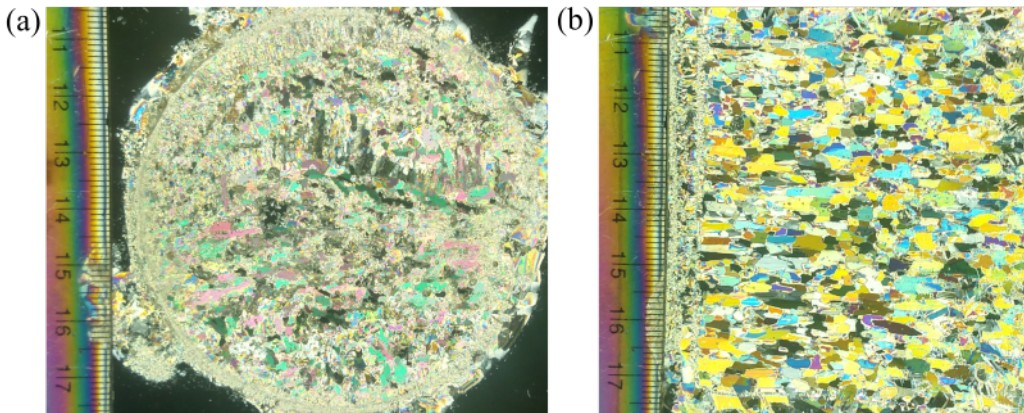

**Figure B1.** (a) Horizontal and (b) vertical thin sections of spray sea ice.

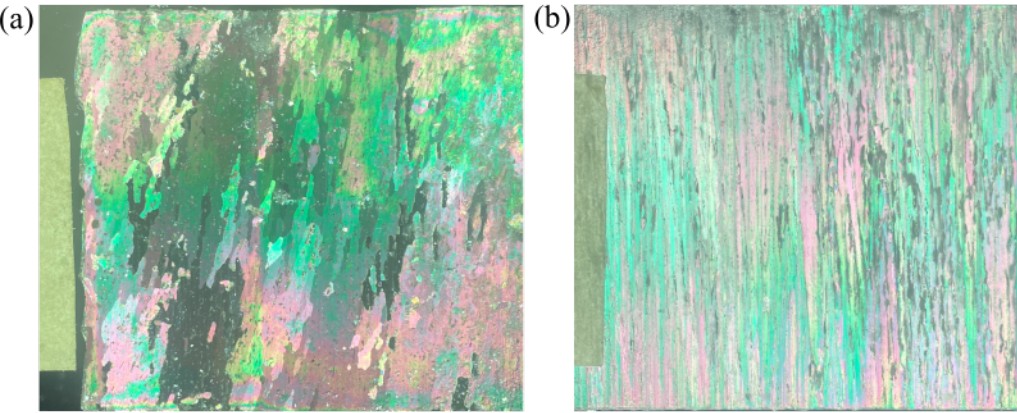

**Figure B2.** (a) Horizontal and (b) vertical thin sections of columnar sea ice.



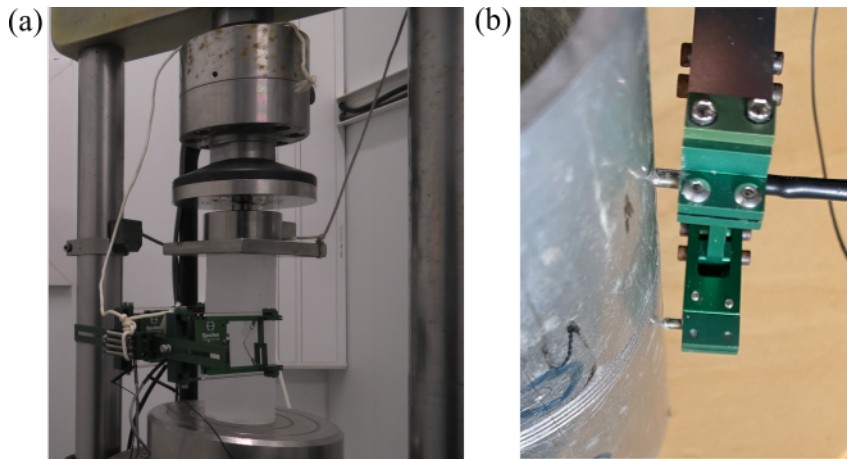

**Figure B3.** (a) Overview of uniaxial compression test. (b) Mounting of strain sensor on a sample.

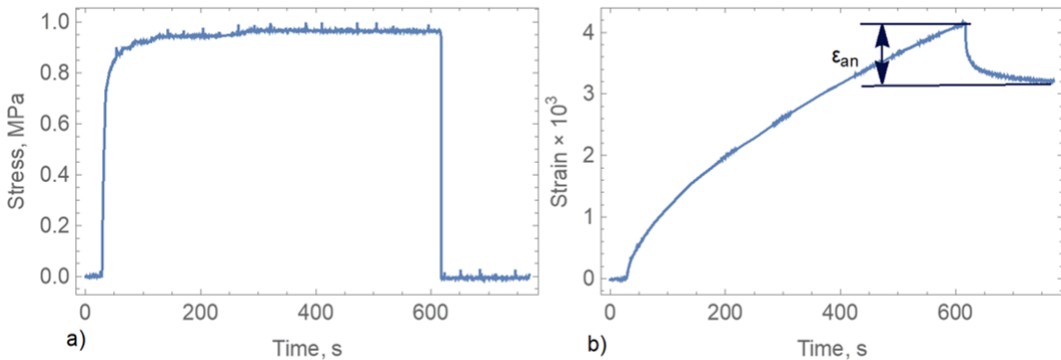

**Figure B4.** Stress and strain versus time in LU test with an ice core of spray ice (ice temperature is 5°C).

tests were performed in the cold laboratory of The University Centre in Svalbard by the test machine Knekkis. The load was measured by two similar HBM load cells 10 T mounted in the rig and placed on the surface of ice core Fig. B3b. Records of the second load cell were synchronized with the records of the EpsilonTech extensometer, with 50mm base, mounted in the middle part of the ice core. Records of the Knekkis load cell were synchronized with the records of vertical displacement of the plate supporting the ice core in the rig.

Deformations recorded by EpsilonTech extensometer were usually smaller the deformations calculated from the records of the displacement sensor in the Knekkis. The difference is explained by ice failure effects at the edges of ice cores. We are sure that strains measured by EpsilonTech sensor reproduces better the strains in the middle part of ice cores which are most important for the description of ice rheology.

    Examples of tests records are shown in Figs. B4–B6. Figures B4a and B5a show records of the stresses versus time in the

tests with spray sea ice and columnar sea ice, respectively. The core of spray sea ice was subjected to one LU test with constant





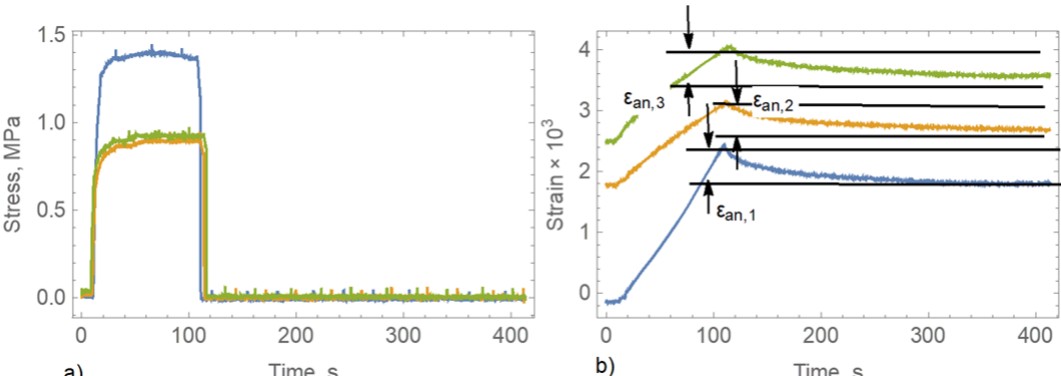

**Figure B5.** Stress and strain versus time in 3 LU tests performed with the same core of columnar ice (ice temperature is 5°C).

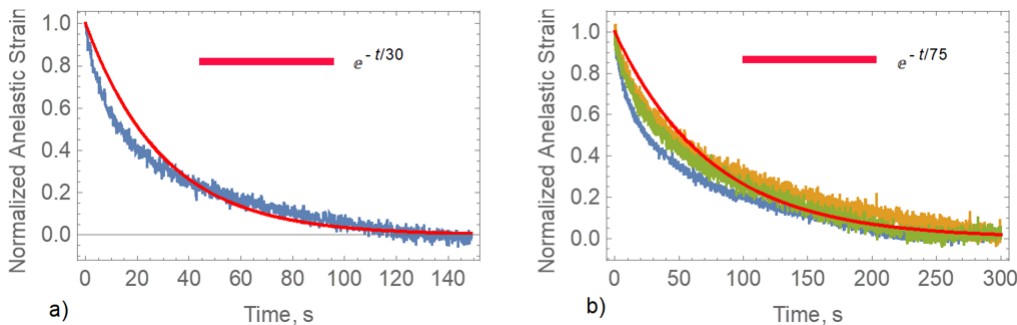

**Figure B6.** Normalized strain versus time in LU tests performed with (a) spray and (b) columnar sea ice cores (ice temperature is 5°C).

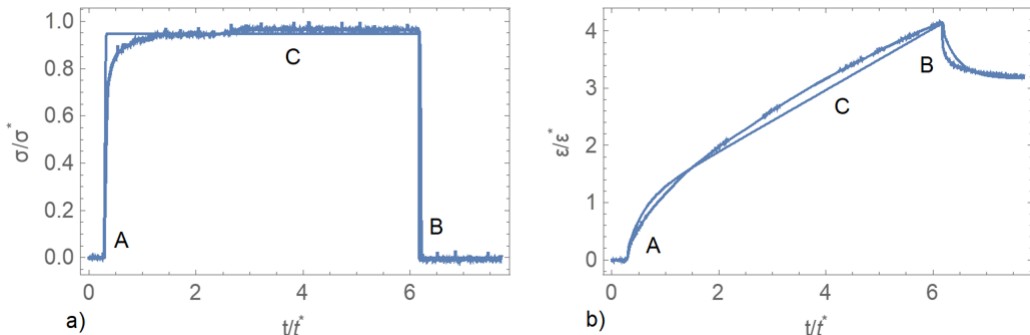

**Figure B7.** (a) Dimensionless stress and (b) normalized strains versus dimensionless time. Thick and thin lines correspond to measured and simulated quantities, respectively. LU test with a core of spray sea ice.



The Cryosphere
Discussions

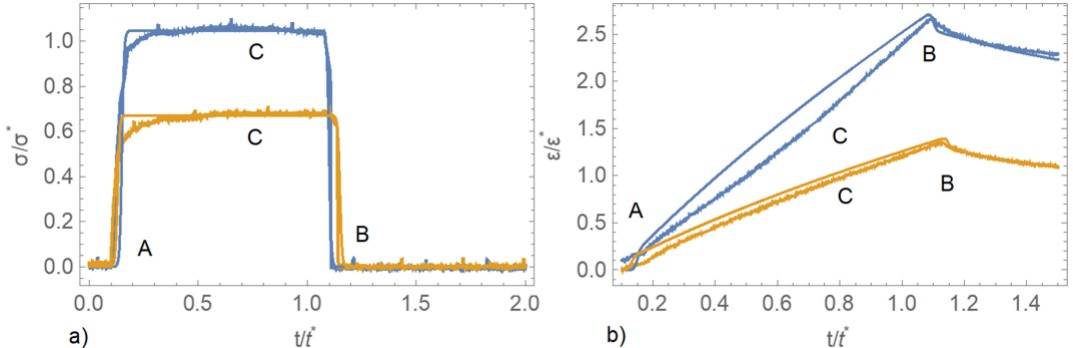

**Figure B8.** (a) Dimensionless stress and (b) normalized strains versus dimensionless time. Thick and thin lines correspond to measured and simulated quantities, respectively. Blue and yellow lines correspond to two LU tests with a core of columnar sea ice.

compression of about 1 MPa for 10 minutes. The horizontal core of columnar sea ice was subjected to three consequent LU tests. Duration of each compression was 100 s. In test 1 the stress was of about 1.5 MPa, and in the next tests 2 and 3 the stress was of about 1 MPa. Blue line, yellow and green lines correspond to the first, the second and the third tests. Figures B4b and B5b show records of the strains versus time in the tests. The return of strains after the loads are removed are well visible in the

figures. Figure B5b shows accumulation of irreversible strains after each test: the initial strain equals zero on the blue line, the initial strain is higher at the yellow line, and the initial strain is maximal at the green line. Figure B6 shows normalized strains in the tests versus time after the load is removed. Representative times of the return of delayed strains are estimated 30 s in the spray ice and 75 s in the columnar ice.

**B2    Rheological constants of spray and columnar ice**

It is assumed that ice rheology is described by Burgers model consisting of linear combination of Maxwell and Kelvin/Voigt units

$$\ddot{\epsilon}\tau_2 + \dot{\epsilon} = \frac{\dot{\sigma}}{E_{eff}} + \ddot{\sigma}\frac{\tau_2}{E_1} + \frac{\sigma}{\eta_1} \tag{B1}$$

where $\sigma$ and $\epsilon$ are stress and strain, and dots above the letters means the time derivatives. Rheological constants are determined by $\tau_2 = \eta_2/E_2$ and $1/E_{eff} = 1/E_1 + \tau_2/\mu$. Here, $E_1$ and $\eta_1$ are elastic and viscous constants of Maxwell unit, and $E_2$

and $\eta_2$ are elastic and viscous constants of Voigt unit.

Acoustic measurements of the speed of longitudinal waves combined with measurements of the natural frequencies of ice beams and discs show that elastic modulus $E_1 \approx 5$ GPa for sea spray ice and for columnar sea ice in the horizontal direction at -5°C (Marchenko et al., 2021). From Fig. B6 it follows that $\tau_2 = 30$ s for sea spray ice, and $\tau_2 = 75$ s for columnar sea ice. The rheological constants $E_2$, $\eta_1$ and $\eta_2$ are found from the approximation of the dependencies shown in Figs. B4b and B5b

by the solution of Eq. B1 obtained with prescribed dependencies of the stress $\sigma$ from the time shown in Figs. B7 and B8 by





**Table B1.** Rheological coefficients of spray sea ice and columnar sea ice.

|  | $E_1$ (GPa) | $E_2$ (GPa) | $\eta_1$ (GPa·s) | $\eta_2$ (GPa·s) | $\mu$ (GPa·s) | $E_{eff}$ (GPa) |
|---|---|---|---|---|---|---|
| **Spray ice** | 5 | 1.2 | 176 | 36 | 30 | 0.8 |
| **Columnar ice** | 5 | 1 | 55 | 75 | 32 | 0.4 |
|  | 5 | 1 | 83 | 75 | 39 | 0.5 |

thin lines. They are given by the equation

$$\frac{\sigma}{\sigma^*} = 0.5 \left( 1 + \tanh\left( 100 \left( \frac{t}{t_*} - t_A \right) \right) \right) \frac{\sigma(t_C)}{\sigma^*} - 0.5 \left( 1 + \tanh\left( 100 \left( \frac{t}{t_*} - t_B \right) \right) \right) \frac{\sigma(t_C)}{\sigma^*} \tag{B2}$$

where the values $t_a$, $t_b$ and $t_c$ correspond to the points A, B, and C shown in Figs. B7a and B8a. Eq. B2 was adjusted to the stress record in each test. Results of tests 2 and 3 performed with a core of columnar sea ice are very similar. Therefore, only
tests 1 and 2 were simulated.

Numerical simulations were performed with dimensionless equation (1) derived with representative stress $\sigma^* = 1$ MPa, strain $\epsilon^* = 0.001$, and $t_* = 100$ s. The initial conditions are $\epsilon(0) = \dot{\epsilon}(0) = 0$. The strains calculated from Eq. B1 with the values of rheological constants shown in Table B1 are shown by thin lines in Figs. B7b and B8b. The thin lines approximate well the recorded strains on the segments ACB.

## Appendix C: Models

An overview of the different wave dispersion and dissipation models used in this study is provided here in order as presented in Fig. 5. For brevity, models are presented with limited context and the reader is referred to the model references for further details.

### C1   Squire and Allan (1977)

The simplified viscoelastic model of Squire and Allan (1977) can be written as (Li et al., 2015; Sree et al., 2018):

$$(\omega^2 - Qgk \tanh kH) = 0 \tag{C1}$$

$$Q = \frac{(G - i\omega\rho_i v)h^3}{6g\rho_w(1-\theta)}k^4 - \frac{\rho_i h \omega^2}{\rho_w g} + 1 \tag{C2}$$

where $\theta$ is the Poisson ratio, here taken as 0.3, and the shear modulus $G = Y/(2(1+\theta))$. We note that $k$ represents the complex
wave number, $k = k_r + ik_i$, and $\alpha = 2k_i$.

### C2   Marchenko et al. (2020)

$$\omega^2 = k \tanh(kH) \left( g + \frac{E_1 h^3 k^4}{12\rho_w} \right) \tag{C3}$$





$$\alpha = \frac{E_1 h^3 k^4}{24 \rho_w \left(g + E_1 h^3 k^4 / 12 \rho_w\right)} \frac{E_1}{\mu_i c_g} \tag{C4}$$

where $\mu_i$ is the ice viscosity. In this study, we used $E_1 = Y$.

### C3 Sutherland et al. (2019)

$$\alpha = \Delta_0 \epsilon h k^2 \tag{C5}$$

where in this study $\Delta_0 = 1$, i.e., a no-slip boundary condition. For a continuous ice sheet, it is expected that $\epsilon$ is small and thus we choose here for $\epsilon = 0.01$. For an ice thickness of $h = 0.35$ m, this corresponds to a highly viscous layer of 3.5 mm.

### C4 Marchenko and Cole (2017)

The spatial attenuation rate is calculated according to the model of Marchenko and Cole (2017) by the formula

$$\alpha = \frac{Y^2 k^5 \tanh(kH)}{4 c_g \rho_w \mu_w \omega^2 (1 - \theta^2)^2} \int_0^h \frac{K}{\phi} dz \tag{C6}$$

where $K$ is the permeability of the ice in m$^2$, $\mu_w$ is the dynamic viscosity of the brine (taken here as $1.5 \times 10^{-3}$ Pa·s), and $\phi$ is the liquid brine volume content calculated with the formula $\phi = \sigma_{si}(49.185/|T| + 0.532)$ (Frankenstein and Garner, 1967) where $\sigma_{si}$ is the sea ice salinity, here taken as 10 ppt. Sea ice permeability is estimate by the formula $K = K_0 \exp(15\sqrt{\phi})$ (Zhu et al., 2006). We estimate $\alpha$ assuming that the sea ice temperature varies linearly from -2°C at the ice bottom ($z = 0$) to -25° at the ice surface ($z = h$) according to $T = -2 - 23z/h$.

### C5 Liu and Mollo-Christensen (1988)

$$\omega^2 = \frac{g k_r + B k_r^5}{\coth(k_r H) + k_r M} \tag{C7}$$

$$c_g = \frac{g + (5 + 4k_r M) B k_r^4}{2\omega(1 + k_r M)^2} \tag{C8}$$

$$\alpha = \frac{\sqrt{\upsilon \omega} k_r}{c_g \sqrt{2}(1 + k_r M)} \tag{C9}$$

with $B = Y h^3 / \rho_w 12(1 - \theta^2)$ and $M = h \rho_i / \rho_w$, and where $g$ is the gravitational acceleration, $H$ is the water depth, $Y$ is the elastic modulus of the ice, $\rho_w$ and $\rho_i$ are the densities of water and ice, respectively, $\theta$ is the Poisson ratio and $h$ is the ice thickness. Note that the contribution of ice compression has been ignored here.





## C6 Kohout et al. (2011)

$$\alpha = 2H_s C_D k^2 \tag{C10}$$

where $C_D$ is the under-ice drag coefficient.

## C7 Stopa et al. (2016)

Considering a fully turbulent boundary layer under the ice and assuming the analogy between the wave-ice boundary layer and the wave bottom boundary layer holds, the wave attenuation rate may be given by:

$$\alpha = f_w \frac{\omega^2 u_{orb}}{g c_g} \tag{C11}$$

where $u_{orb} = \omega a_0$ is the wave orbital motion, $a$ is the wave amplitude and taken as $H_s/2$, and $f_w$ is the friction factor for which we use here the simplified model of Soulsby (1997):

$$f_w = 1.39 \left( \frac{a}{k_s/30} \right)^{-0.52} \tag{C12}$$

$$f_{w,max} = 0.3 \tag{C13}$$

Here, $k_s$ is the Nikuradse roughness height. We refer to Stopa et al. (2016) the WaveWatchIII source code for more details on Eq. C11.

*Author contributions.* JV and AB conceptualized. JV and JR constructed instrumentation. KF and IR deployed and retrieved instrumentation. AM retrieved ice cores from the Arctic, designed and performed laboratory experiments on ice cores. QL and JL assisted in the modeling. AB and AM administered the project. JV and AM prepared the original draft. JV, QL, AM, JR, PH, TW, TN, TK discussed results and edited the manuscript. All authors reviewed the manuscript.

*Competing interests.* The authors declare that there is no conflict of interest.

*Acknowledgements.* We acknowledge the use of imagery from the NASA Worldview application (https://worldview.earthdata.nasa. gov/, last access: 17 April 2021), part of the NASA Earth Observing System Data and Information System (EOSDIS). Authors would like to thank Flynn Jackman and the rest of the team at Casey Station, Antarctica, for their assistance in deploying instrumentation and collecting ice samples. Data collection in Grønfjorden, Svalbard, was conducted within the expedition "Spitsbergen-2020" organized by the Russian Scientific Arctic Expedition on Spitsbergen Archipelago (RAE-S), AARI. Joey J. Voermans, Qingxiang Liu and Alexander V. Babanin





acknowledge support from the Joyce Lambert Antarctic Research Fund and the U.S. Office of Naval Research Grant N62909-20-1-2080.
Joey J. Voermans, Alexander V. Babanin and Petra Heil were supported by the Australian Antarctic Program under project 4593, and Petra
Heil under project 4506. Joey J. Voermans, Jean Rabault, Kirill Filchuk, Aleksey Marchenko, Takuji Waseda, and Alexander V. Babanin
acknowledge the support of the Research Council of Norway through the SFI SIB project. Jean Rabault was supported in the context of the
DOFI project (University of Oslo, grant no. 280625).



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
