# Peer review of "Wave dispersion and dissipation in landfast ice: comparison of observations against models"

_The Cryosphere, 2021_

## Author Comment (AC1)

**Review #1**

This is a well-written manuscript with a thorough analysis on a set of difficult to obtain field data. The topic is of significant climate interest. This reviewer strongly supports its publication.

All comments below are on minor issues, but should be addressed before publication.

1. Proofread the whole manuscript to eliminate some typos. For example: line 54, remove "be"; line 335, change "smaller the" to "smaller then".

> **We thank the reviewer for pointing this out. We confirm that the manuscript has been proofread again and typos have been corrected.**

2. Questions on some reference.

2a. The reviewer could not find the spectral attenuation data from Thomson et al. 2018 as mentioned in Fig. 4. The figure caption says that these data were in Liu et al 2020. However, in the Liu et al paper (Fig. 10) one clearly sees 10(c) corresponds to the pancake group in the present manuscript, and 10(f) corresponds to the broken pack ice group in the present manuscript, but no where can the grease ice data be found in Liu et al 2020. Please clarify or revise since this is an important dataset for many readers.

> **The reviewer is correct that the data used in Fig. 4 cannot be directly obtained from Thomson et al. (2018) who published only the non-averaged values of their experiment. Instead, in the comparison we printed the experiment averaged values which were retrieved from Liu et al. (2020) which are based on the data published in Thomson et al. (2018). Experimental averages of the observations in grease ice are provided explicitly in Kodaira et al. (2020). We note that this is indicated in the caption of Fig. 4, but acknowledge that this was not clarified in the main text. We will therefore mention this in the main text as well.**

2b. Line 45, the reference Liu et al 2020 is on comparison of different Sice models, not on model calibration. There are two other papers both exactly on calibration: one using in-situ data and the other using satellite observations. Replacing Liu et al 2020 by these two (and maybe other too) is appropriate: Cheng, S., Erick Rogers, W., Thomson, J., Smith, M., Doble, M. J., Wadhams, P., . . . Shen, H. H. (2017). Calibrating a viscoelastic sea ice model for wave propagation in the Arctic fall marginal ice zone. Journal of Geophysical Research: Oceans, 122, 8770–8793. https://doi.org/10.1002/2017JC013275 and Cheng, S., Stopa, J., Ardhuin, F., and Shen, H. H.: Spectral attenuation of ocean waves in pack ice and its application in calibrating viscoelastic wave-in-ice models, The Cryosphere, 14, 2053–2069, https://doi.org/10.5194/tc-14-2053-2020, 2020.

> **On the suggestion of the reviewer, we will replace the reference of Liu et al. (2020) by Cheng et al. (2017) and Cheng et al. (2020).**

2c. Line 57. The following paper is on ice tongue measurements. It should also be included as in situ data for landfast ice: Squire, V., Robinson, W., Meylan, M., & Haskell, T. (1994). Observations of flexural waves on the Erebus Ice Tongue, McMurdo Sound, Antarctica, and nearby sea ice. Journal of Glaciology, 40(135), 377-385. doi:10.3189/S0022143000007462.

**We thank the reviewer for the reference, and will include it in the manuscript.**

3. In section 2, for completeness, a brief description of the meteorological conditions would be helpful.

**Unfortunately, no wind atmospheric conditions were measured by the instrumentation deployed. However, an indication of the atmospheric conditions can be retrieved from a nearby weather station at Isfjorden. During the experiments, the wind speed ranged from 0-22 m/s, with a mean of about 8 m/s. We note that this wind speed is somewhat higher than what we would expect in Gronfjorden. We will add this information in the manuscript.**

4. Appendix C. All models here have alpha explicitly given except C1. For completeness the authors should also obtain alpha from Eqs. (C1&C2) and provide it here.

**For this model, retrieving alpha from Eq. C2 is not trivial. For completeness, we already defined $\alpha = 2k_i$ included in-line below C2 but will, on the suggestion of the reviewer, number this formula explicitly as well.**

5. Lines 112-113. "For the Antarctic experiment, this assumption was tested using ERA5 re-analysis data in the open water just north of the marginal ice zone indicating a relative bearing of approximately 15 degree." What does this mean? A 15 degree off the line between the buoy pair? How was this accounted for? Did the authors modify the distance in Eq. (3) accordingly? In fact, since each frequency might have its own direction, in fact a directional spectrum is needed. Can the authors discuss this in the manuscript?

**Indeed, the incoming waves are not perfectly aligned with the transect of the buoy pair in the Antarctic experiment. Unfortunately, no reliable information is available to determine the direction of waves, for which we therefore have to rely on ERA5 re-analysis. We did not correct for this difference, as we cannot guarantee the accuracy of ERA5. Nevertheless, a 15 degree bearing would imply an error (underestimation) of about 5% of the estimated attenuation rates, which is smaller than the uncertainty of the Antarctic observations.**

**To attend the reader on this limitation, we will add the following to the manuscript:** *"As we do not have in-situ observations of the wave directions (ideally obtained through the directional wave spectrum), we did not correct for this misalignment. We note that for a relative bearing of 15° the estimated attenuation rate would be increased by no more than 5%."*

6. Lines 122-126. The authors remind us that ice viscosity is a parameter, which value depends on the spring-dashpot model used. Hence, it is also necessary to remind us which model was used in Tabata 1958 and Lindgren 1986 when they reported their measured ice viscosity values. The model Marchenko et al 2020, 2021 used is already given in the present manuscript.

**Tabata (1958) and Lindgren (1986) describe the ice as a Burgers material (Maxwell-Voigt spring-dashpot model). We will add this information in the manuscript.**

7. Lines 219-222. This discussion on the source of turbulence triggers a question: what is the tidal condition in the fjord?

At Gronfjorden, where the Arctic experiment was performed, the maximum tidal range was approximately 1.5 m. However, while this may be a source of turbulence under the ice, it is unlikely to have contributed to attenuation of waves, at least, to a first order approximation.

---

## Author Comment (AC2)

**Review #2**

The manuscript „Wave dispersion and dissipation in landfast ice: comparison of observations against models" by Joey Voermans and colleagues is devoted to the analysis of wave propagation and dissipation in landfast sea ice. The study is based on data from field measurements (with IMUs placed on the ice) from two locations, one in the Arctic and one in the Antarctic. The observations are used to estimate the wave numbers and attenuation coefficients of waves within the frequency range of approx. 0.05–0.2 Hz. The obtained attenuation coefficients are compared with those predicted by several models of wave energy dissipation within sea ice and in the turbulent boundary layer under the ice.

Undoubtedly, the problems discussed in the study are important for the current research on sea ice– wave interactions. Our better understanding of the physical mechanisms underlying wave energy attenuation in sea ice is crucial for developing better parameterizations of those processes and thus for a better performance of local and larger-scale sea ice models. Moreover, in my opinion the manuscript is well written, the presented analysis thorough and convincing, and the discussion interesting. My  recommendation is to accept the manuscript for publication in The Cryosphere after the Authors address the comments listed below.

Major comments:

1. I really like the introduction: it is relatively short, but well formulated and contains all relevant information needed as a background for the results presented in the manuscript.

   Just one comment to that part: I'd suggest stating explicitly that equation (1) is based on an assumption that the dissipative processes leading to wave energy attenuation are linear. This assumption is not always true (see, e.g., Squire, Phil. Trans. A, 2018 or Herman, J. Phys. Oceanogr., 2021), and considering that the present study identifies turbulence as an important source of energy dissipation, and that turbulence is strongly nonlinear, I think it is worth mentioning already in the introduction (the Authors state it later, in lines 105-106, when equation (3) is introduced).

   Note that the models of Kohout et al. 2011 and Stopa et al. 2016 both produce non-exponential attenuation rates (see equation 12 in Kohout et al. 2011; analogously, the coefficient β in equation B1 of Stopa et al. is a function of wave energy – for high Reynold numbers, the model is analogous to the bottom friction formulation in spectral wave models, which is nonlinear).

   **We agree with the reviewer. There is the implicit assumption in equations (1) and (3) that the dissipative processes responsible for wave attenuation are linear. As the reviewer mentions, we acknowledge this indirectly at a few instants throughout the manuscript, but not explicitly. We will therefore add the following just before Eq. (1):** *"when wave scattering is assumed to be insignificant and wave dissipative processes are approximated as linear"*. **Moreover, just after Eq. (3), we will add the following: "***Eq. (3) implicitly assumes that the dissipative processes are linear, i.e., the rate of energy dissipation is independent of the wave energy or wave amplitude (e.g., Squire at al, 2018). Evidently, this is not the case for all dissipative processes, such as turbulence (e.g. Herman, 2021; Voermans et al., 2020; Stopa et al., 2016; Kohout et al., 2011). Nevertheless, due the*

*simplicity of Eq. (3) and the experimental restrictions of our field experiments we will use Eq. (3) in determining the wave attenuation rate".*

2.  On page 5 the Authors write: "We note that for the Arctic experiment only those observations are used that were obtained from the buoy pair furthest apart as they were deemed most accurate." This statement implicitly means that the Authors assume that the attenuation coefficients computed from their data are distance dependent – buoys placed 600 m and 800 m apart are expected to produce different results than those placed 1400 m apart. But then why are 1400 m enough? Can we expect attenuation coefficients obtained for larger buoy distances to be different? How? Do they depend on buoy-buoy distance in a systematic way? Are the differences between attenuation coefficients computed for different distances wave-frequency dependent and do they thus affect the resulting slope of alpha(f) – which is the main result of this paper?

    Did the Authors compute α from buoys 1-2 and 2-3 and compare the results with those for buoys 1-3, presented in the paper?

    It must be also remembered that, for the longest waves considered in the analysis (f<0.1Hz), even buoys 1-3 are only ~3 wavelengths apart, i.e., the distance over which attenuation is measured can be regarded as very short. Do the Authors agree that this fact might have some influence on the results? (Please note that I'm not criticizing the fact that the buoys were placed the way they were – there might have been several practical reasons for that – but only that the Authors don't pay any attention to the possible role of buoy-buoy distance in their analysis).

    Overall, considering how limited the dataset is (9 and 2 data points for the Arctic and Antarctic), I'd say that data from 3 buoy pairs are better than from just one!

    **We thank the reviewer for the feedback. The main reason we presented the observations for the buoy-pair that is furthest apart only is because the scatter is significantly larger for the other two buoy pairs. The reason for this large scatter is very simple: the shorter the distance between the buoys, the harder it becomes to accurately measure wave attenuation, as it scales with the distance of wave propagation.**

    **Based on the comment of the reviewer, we looked again at the attenuation rates between the other two buoy pairs, see Figure below. As expected, the scatter between those buoy pairs with the shortest separation distance (top two plots) is significantly larger than for the buoy-pair with the largest separation distance (bottom left). The vertical dashed line refers to the wave period at which the wave length is about 3 times the separation distance.**

    **If one would ignore the uncertainties in $\alpha$ for buoy-pairs 1-2 and 2-3, for the sake of discussion, we may observe that the wave attenuation rates are indeed largest for the buoy pair closest to the ice edge (where wave energy is largest), and wave attenuation rates are smallest for the buoy pair furthest from the ice edge (where wave energy is lowest). The attenuation between the pair furthest apart is, as one would expect, somewhere in between. This is consistent with comment (1) of the reviewer, and thus would substantiate that the dissipative process is non-linear. We note that, for $T > 12$ s**

**no reliable data of $\alpha$ can be obtained for buoy-pairs 1-2 and 2-3 as the distance between the buoys is too short for significant dissipation to occur. Interestingly, we see that the slope of $\alpha(f)$ increases slightly with distance from the ice edge, at least, if Eq (3) is used in determining $\alpha$.**

**However, we need to be very careful in interpreting the estimated attenuatuion rates from buoy-pairs 1-2 and 2-3 as the uncertainty in $\alpha$ for these pairs is similar to the differences in the mean values of $\alpha$ between the different pairs. Nevertheless, given the rarity of the observations we will add the figure below in the Discussion section, including an additional paragraph describing the observations and its limitations.**

[Figure]

3. Related to the previous comment: I'm wondering whether the Authors have any information on the open water wave heights corresponding to those measured within the ice. For the Antarctic experiment, ERA5 data are mentioned (line 112), but also the fact that ice pack was present between the open ocean and the fastice, which certainly modified the wave energy reaching the fastice edge. What about the Arctic experiment?

What I mean is: It would be interesting to see how the open water spectra computed from those measured in the ice and from the computed attenuation rates compare with the corresponding open water spectra from spectral models or other sources.

**We agree with the reviewer that this would be interesting to look at. In fact, during the Arctic measurement campaign we deployed a wave buoy at the entrance of Gronfjorden to measure the incoming wave field. Unfortunately, one week into the experiment (before the ice buoys were deployed on the ice), the wave buoy mooring was dragged from its deployment location by sea ice and a few days after no Iridium transmissions were received anymore. We agree that hindcasts would be of interest, however, considering the huge efforts required to do so (bathymetry and fjord system is very complex), we leave that for future research.**

4. Discussion, lines 213-217: the model of Liu and Mollo-Christensen (1988) is suitable only for a laminar boundary layer! It is simply incorrect to try to make it suitable for turbulent boundary layers by increasing the viscosity as much as one finds it necessary in order to make the model fit the observations – although several numerical studies do exactly that. The viscosity in the model of Liu and Mollo-Christensen (1988) has a physical meaning and cannot be simply increased by a few orders of magnitude when that seems necessary. Crucially, in a laminar boundary layer the viscosity does not depend on wave energy, but in a turbulent boundary layer it does. Difficulty with calibrating the models to perform well in both calm and storm conditions is one of the consequences of the (mis)use of the Liu and Mollo-Christensen (1988) model for turbulent dissipation – as analyzed e.g. by Li et al. 2015.

**We thank the reviewer for the comments, which made us look at the argument from a reader's point of view. Indeed, Liu and Mollo-Christensen (1988) use a viscous argument, and is thus based on different physics as the kinematic viscosity is a fluid property (as opposed as the eddy viscosity, which is a flow property). In this regard, we used their approach by analogy, that is, similar to the viscous stress, the turbulent stress is proportional to the product of the velocity gradient and the eddy viscosity. Unlike the molecular viscosity, the eddy viscosity is a dynamic property and thus may depend on the waves too (something we also acknowledged in the manuscript, Lines 222-225). We note that this is the traditional way of introducing turbulent viscosity, including in the paper of Liu and Mollo-Christensen (1988, see their Appendix). We will make this distinction clear in the updated manuscript through the addition of the following in the Results section:** *"While common, caution is required in replacing the molecular viscosity by an effective viscosity as the physical problem stipulated by Liu and Mollo-Christensen (1988) considers a laminar boundary layer under the ice only".*

Minor (mostly technical) comments:

1. Page 3, line 54: "is never be"

   **Corrected**

2. Page 3, line 57: Why "perhaps"?

**Removed**

3. Page 4, line 91: omega has not been defined (and f is used instead of omega throughout the paper)

   **We have added the definition of $\omega$.**

4. Page 5, line 113: "indicating a relative bearing of approximately 15°". What does "relative bearing" exactly mean? The angle between wave propagation and the line connecting the two buoys? And were its value constant throughout the whole period of the experiment?

   **The relative bearing is indeed the angle between the line of buoys and the wave direction. Based on ERA5 re-analysis, it is reasonably constant throughout the experiment.**

5. Page 6, line 135: $\rho$ hasn't been defined.

   **We have added the definition of $\rho$.**

6. Figure 2, measured data: what exactly do circles, bars and vertical lines mean? (i.e. standard deviations or percentiles, etc.?)

   **These are percentiles. Here, vertical lines are the 9th and 91st percentiles, and boxes are 25th and 75th percentiles. The circles are mean values. We have added this in the caption of Figure 2.**

7. Figure 3a: I'm not sure if I can see it correctly, but there are crosses inside of some of the circle symbols – please clarify.

   **Crosses identify the mean values. This is the same as the circles, we therefore removed the crosses.**

8. I find it inconsistent that the data in Figs. A2 and 2 (with the corresponding text in the first part of the Results section) are presented in terms of wave frequency, and then the data in Figs. 3–5 is plotted and discussed in the text in terms of wave periods. It's not wrong, of course, and the Authors may decide to leave it the way it is if they prefer to do so, but it makes comparisons between plots less easy.

   **We agree with the reviewer that comparison between the figures may have been complicated by the current presentation. Our preference is plotting against wave period. Unfortunately, plotting Figure 2 in terms of wave period (as opposed to wave frequency) makes the data difficult to read, even at log-scales. We therefore decided to keep the axes unchanged, but have included an additional horizontal axis to Figure 2 at the top to explicitly state the corresponding wave period values.**